# The Effects of the Marination Process with Different Vinegar Varieties on Various Quality Criteria and Heterocyclic Aromatic Amine Formation in Beef Steak

**DOI:** 10.3390/foods11203251

**Published:** 2022-10-18

**Authors:** Halenur Fencioglu, Emel Oz, Sadettin Turhan, Charalampos Proestos, Fatih Oz

**Affiliations:** 1Department of Food Engineering, Agriculture Faculty, Atatürk University, 25240 Erzurum, Türkiye; 2Department of Food Engineering, Faculty of Engineering, Ondokuz Mayıs University, 55139 Samsun, Türkiye; 3Laboratory of Food Chemistry, Department of Chemistry, National and Kapodistrian University of Athens, Zografou, 15771 Athens, Greece

**Keywords:** vinegar, marination, texture, SDS-PAGE, heterocyclic aromatic amines

## Abstract

Herein, the effect of the dipping (static) marination process (at 4 °C for 2 h) with different types of vinegar (balsamic, pomegranate, apple, and grape) on various quality properties, including texture and protein profile of beef steaks and the formation of heterocyclic aromatic amines (HAAs) in beef steaks cooked on a hot plate (at 200 °C for 24 min), were determined. The results showed that 3.12–4.13% of the marinate liquids were absorbed by beef steak as a result of the marination process. No significant differences (*p* > 0.05) were observed between the marinated and cooked beef steaks in terms of water content, cooking loss, thiobarbituric acid reactive substances (TBARS) value, hardness, cohesiveness, and chewiness. However, significant differences were detected in terms of pH value and color values (L*, a*, and b*) (*p* < 0.01), and springiness, 2-amino-3,8-dimethylimidazo (4,5-f) quinoxaline (MeIQx) and total HAA content (*p* < 0.05). The marination with pomegranate vinegar resulted in the formation of darker steak, while a lighter one was obtained when apple vinegar was used in the marination. The use of balsamic and grape vinegar in the marination process decreased the springiness value compared to the control group. The myofibrillar proteins of beef steaks marinated with different types of vinegar generally showed a similar sodium dodecyl-sulfate polyacrylamide gel electrophoresis (SDS-PAGE) profile. However, some differences were observed in the band density of some proteins depending on the trial and the type of marination. In this study, of the nine examined HAAs, only two (2-amino-3-methylimidazo (4,5-f) quinoline (IQ) and MeIQx) could be detected and quantified. IQ was detected only in the control group steak (up to 0.51 ng/g), while MeIQx was detected in all treatment groups (up to 2.22 ng/g). The total HAA content varied between 0.59–2.22 ng/g. It was determined that the marination process with different vinegar types had different effects on the total HAA content of the steaks. Using balsamic and apple vinegar in the marination process decreased the total HAA content compared to the control group, but this decrease was not statistically significant (*p* > 0.05). On the other hand, using grape and pomegranate vinegar in the marination process increased the total HAA content, but this increase was only significant (*p* < 0.05) in the marination with pomegranate vinegar.

## 1. Introduction

Meat is a nutrient-dense food with high nutritional value due to its protein, fat, vitamins, and minerals, which are significant for good nutrition. Quality has increasingly gained prominence in meat and meat products, as well as other foods, due to consumer awareness. Meat quality is defined by various characteristics such as color, taste, juiciness, and texture [1] and is affected by various factors during the pre-rigor, rigor, and post-rigor phases [2].

In general, maturation refers to the transformation of muscle into meat and the creation of the desired textural properties by consumers. This can be done more quickly using different techniques and can occur naturally to a certain extent by storing the meat. In this context, marination is a widely used process to improve the sensory and textural properties of meat [3]. In the marinating process, salts, organic acids, fruits and vegetables, drinks (soda, wine, etc.), enzymes, and vinegar, as well as combinations of these agents, are commonly used as marinades. It has been demonstrated in various studies that with marination, the water-holding capacity of the product increases and the cooking loss decreases. As a result of the decrease in muscle fibril hardness, the textural properties improve, and the color, taste, and aroma properties are developed. Antioxidant ingredients reduce flavor losses, antimicrobial ingredients limit microbial growth, and the shelf life of the product is extended [4,5,6,7,8,9].

Vinegar is one of the most common marinades for acidic marination. Vinegar is defined as a fermented food product obtained by subjecting mostly grains and fruits to alcohol and then acetic acid fermentation [10,11]. Fruits with high sugar content are needed for vinegar production. Various fruits, especially grapes, apples, figs, pomegranates, raspberries, mulberries, dates, coconut, sour cherries, pears, and plums, are used in the manufacture of fruit vinegar [12,13], and vinegar is referred to as the fruit used in its production. Balsamic vinegar is typically made by aging boiled grape juice in various barrels (grape or oak) for an extended period [14]. Vinegar, a valuable fermentation product because of its bioactive compounds such as organic acids, amino acids, phenolic compounds, and melanoidins, is commonly used in the food industry and kitchens due to its antimicrobial, antioxidant, antidiabetic, anticarcinogenic, and anti-infection properties [11].

Meat, except for products such as sushi and pastırma (a Turkish-type dried meat) that are eaten without heat treatment, is consumed after cooking. The cooking process has a significant impact on the tenderness, juiciness, and flavor of the meat. During the cooking process, changes in protein, which is one of the most significant nutritional benefits of meat, are one of the key factors that determine the quality of cooked meat. Cooking improves digestibility by denaturing proteins, producing a tastier and more stable product [15,16]. The denaturation of proteins, on the other hand, reduces the meat’s water-holding capability. As a result, the water removed from the food during the cooking process often removes certain water-soluble substances. Furthermore, various losses in vitamins and mineral substances that are not heat resistant can be observed [16,17,18]. Moreover, certain food toxicants, such as heterocyclic aromatic amines, may be produced while cooking protein-rich foods like meat.

Heterocyclic aromatic amines (HAAs) are carcinogenic and/or mutagenic compounds that can form during the heat treatment of protein-rich foods [19,20]. Until today, approximately 30 HAAs have been isolated and identified from foods, according to studies [21]. HAAs, first discovered in grilled meat and fish in 1977, have been divided into two chemical groups. The first group, the aminoimidazoazoarenes, are HAAs that can form at cooking temperatures below 300 °C. The reaction between hexoses, creatine/creatinine, and free amino acids produces these compounds [22]. Aminocarbolines, the other group of HAA, are HAAs that can be produced at temperatures higher than 300 °C. The pyrolysis of amino acids results in the formation of these compounds [23]. It was reported that HAAs are 100 times more mutagenic than aflatoxin B1 and 2000 times more mutagenic than benzo[a]pyrene [24]. Furthermore, it is claimed that there is a beneficial association between different cancer forms and meat consumption and that HAAs played a role in this relationship [25,26]. The International Agency for Research on Cancer (IARC) has classified some of the HAAs as “possible” and “probable” human carcinogens in Groups 2A and 2B as a result of the studies [27]. Since HAAs are carcinogenic and/or mutagenic compounds, recent studies aim to reduce the level of formation of these compounds in heat-treated meat and meat products. Many studies have looked into the effects of garlic [28], red wine [29], green tea [30], black pepper [24], red pepper [31], rosehip [32], chitosan [18], hawthorn [33], cellulose fiber [34], artichoke [35], and basil [36] on HAA formation. However, to the best of our knowledge, no report investigating the impact of vinegar-only on HAA formation in the marination phase has been found in the literature review. Therefore, this study aimed to see how different vinegar varieties (balsamic, pomegranate, apple, and grape) affected HAA formation and meat quality in beef steaks.

## 2. Materials and Methods

### 2.1. Material

The beef steak meat was provided by three two-year-old male beeves (Simmental). At 24 h after slaughter (in post rigor phase), the meats were transported to the laboratory in a chilled state, cut into 2 cm slices, and kept at room temperature for 30 min before the analyses/processes. Balsamic (acidity 5–6%), pomegranate (acidity 4–5%), apple (acidity 4–5%), and grape vinegar (acidity 4–5%) from the same brand were used in the marination process in the current study, and these vinegars were purchased from a local store.

### 2.2. Chemicals

The following chemicals were purchased from Toronto Research Chemicals (Downsview, ON, Canada): 2-amino-3-methylimidazo (4,5-f) quinoline (IQ, CAS no.: 76180-96-6), 2-amino-3-methylimidazo (4,5-f) quinoxaline (IQx, CAS no.: 108354-47-8), 2-amino-3,4-dimethylimidazo (4,5-f) quinoline (MeIQ, CAS no.: 77094-11-2), 2-amino-3,8-dimethylimidazo (4,5-f) quinoxaline (MeIQx, CAS no.: 77500-04-0), 2-amino-3,4,8-trimethylimidazo (4,5-f) quinoxaline (4,8-DiMeIQx, CAS no.: 95896-78-9), 2-amino-3,7,8-trimethylimidazo (4,5-f) quinoxaline (7,8-DiMeIQx, CAS no.: 92180-79-5), 2-amino-1-methyl-6-phenylimidazo (4,5-b) pyridine (PhIP, CAS no.: 105650-23-5), 2-amino-9H-pyrido (2,3-b) indole (AαC, CAS no.: 26148-68-5), 2-amino-3-methyl-9Hpyrido (2,3-b) indole (MeAαC, CAS no.: 68006-83-7), and 2-amino-3,4,7,8-tetramethylimidazo (4,5-f) quinoxaline (4,7,8-TriMeIQx, CAS no.: 132898-07-8, as the internal standard). For the solid phase extraction, an Oasis MCX cartridge (3 cm^3^/60 mg, 30 μm) of Waters (Milford, MA, USA) was used. All the other chemicals were high-performance LC (HPLC) or analytical grade. Water was from a MilliQ water purification system (Millipore, Bedford, MA, USA).

### 2.3. Methods

#### 2.3.1. Marinating Process

Following the delivery of various forms of vinegar used in the marinating process, their acidity levels were assessed. The preliminary testing revealed that the dilution process was needed due to the vinegar’s high acidity. As a result, the vinegar used in the current study’s marination process was diluted to 0.5% acidity with purified water. The sliced beef steaks were separated into 5 pieces of approximately the same dimensions and put into special bags (Elektrola, EVP-SV2230). A randomly selected group was not marinated, this being the control group. To the other 4 groups, the above forms of vinegar (200 mL) were added. Then, all samples, including the control group, were kept at 4 °C for 2 h. Thus, the marinated samples were marinated according to the dipping (static) marination process.

#### 2.3.2. Cooking Process

Meat samples were cooked for a total of 24 min (each side for 12 min) on a hot plate with a surface temperature of 200 °C. To adjust the cooking temperature, a laboratory-type thermometer (Testo 926, Lenzkirch, Germany) was used, and to manage the cooking time a laboratory-type chronometer (EAI, T-590) was used. The steaks were cooked, cooled to room temperature, and weighed. After color analysis, the steaks were homogenized in a kitchen chopper (AL 780, Altus).

#### 2.3.3. Water Content

The water content of raw and cooked beef was determined according to Gokalp et al. [37]. For this aim, the dry matter content was determined by keeping approximately 10 g of the sample in an oven at 100 °C until it reached a constant weight.

#### 2.3.4. Cooking Loss

The weight of the steaks before and after cooking was used to measure the cooking loss values of the samples [38]. Using the difference between the weights, the cooking loss value was calculated as %.

#### 2.3.5. pH Analysis

The pH values of both the vinegars and raw and cooked steak samples were determined using a pH meter (Mettler-Toledo, SevenCompact S210, Greifensee, Switzerland). For the vinegars, direct readings were made using a pH meter. For steak samples, a 10 g sample was homogenized for 1 min in 100 mL distilled water (ultra-turrax, IKA Werk T 25, Staufen, Germany) [37]. Then, the pH values of the steak samples were read using the pH meter. The pH value of the vinegar samples was measured directly. The pH meter was turned on 15 min before use and calibrated with buffer solutions of pH 4.0 and pH 7.0.

#### 2.3.6. Lipid Oxidation

The lipid oxidation level of raw and cooked steaks was determined using thiobarbituric acid reactive substances (TBARS) analysis [39]. Two grams of meat samples were homogenized (ultra-turrax, IKA Werk T 25, Staufen, Germany) for 30 s in 12 mL of trichloroacetic acid solution (7.5% TCA, 0.1% EDTA, 0.1% propyl gallat) and 3 mL of thiobarbituric acid solution (0.02 M) was applied to 3 mL of filtrate after straining through Whatman 1 filter paper. This mixture was kept in a 100 °C water bath for 40 min, then cooled down to room temperature. After centrifuging for 5 min at 2000 rpm, absorbance values against the blank sample at 530 nm were determined in the spectrophotometer. 1,1,3,3-tetraethoxypropane was used for the calculation of value. TBARS values were expressed in mg malonaldehyde/kg.

#### 2.3.7. Color Analysis

The surface color densities of raw and cooked beef steaks were determined using a Minolta color analyzer (CR-400, Minolta Co, Osaka, Japan). The average of four random readings from the samples put on a white background was taken, and the average of the L*, a*, and b* values of the samples were calculated. Color analysis was performed after the temperature of all samples (raw and cooked) was brought to room temperature and before the homogenization process.

#### 2.3.8. Texture Profile Analysis

Textural properties of the cooked meats with a mean of 50 mm diameter and 10 mm thickness were determined at 25 °C according to the methodology described by Öztürk and Turhan [40] using a Texture Analyzer (TA-XT Plus, Stable Micro Systems, Godalming, UK) equipped with an aluminum cylindrical probe (model P/50R). The meats were compressed twice to 30% of their original height at a pre-test speed of 2.0 mm/s, a post-test speed of 5.0 mm/s, a test speed of 5.0 mm/s, and 5 s between compressions. The hardness (N), springiness (mm), cohesiveness, and chewiness (N.mm) parameters of the samples were calculated from the curves provided by the equipment.

#### 2.3.9. SDS-PAGE Profile of Myofibrillar Proteins

The changes in myofibrillar proteins of cooked steaks were determined using the Sodium Dodecyl Sulphate Polyacrylamide Gel Electrophoresis (SDS-PAGE) technique. Myofibrillar proteins were extracted using Molina and Toldra’s [41] procedure with minor modifications [42]. In this context, the samples were homogenized in 30 mM phosphate buffer (pH 7.4) at a 1:10 (*w*/*v*) ratio with ultra-turrax (ultra-turrax, IKA Werk T 25, Staufen, Germany). The homogenate was then centrifuged for 20 min at 10,000× *g* at 4 °C. For the pellets, the supernatant was filtered twice through glass wool. The pellet was then homogenized in 9 volumes of 100 mM phosphate buffer (pH 7.4) and centrifuged for 20 min at 10,000× *g* at 4 °C. The supernatant was collected as a myofibrillar protein fraction and maintained at 4 °C until it was used.

Myofibrillar protein extract concentrations were identified using Bradford reagent (Sigma) and bovine serum albumin (Sigma), and samples were mixed 1:1 (*v*/*v*) with 50 mM tris buffer (pH 6.8). The mixture was held at 100 °C for 5 min before being stored at −18 °C until it was used. A 5% loading gel (pH 6.8) and a 12% separation gel (pH 8.8) were used to separate myofibrillar proteins. 10 µL of each treatment group’s sample was loaded into gel wells. The paint trace was followed until it hit the end of the gel, which was done at 100 V. The gel was stained with a staining solution (Coomassie Brilliant Blue R-250 (1 g/L), 50% (*v*/*v*) methanol, 10% (*v*/*v*) acetic acid) after the electrophoresis (Bio-Rad mini-protean tetra system) process was completed. Following the dyeing process, the dye was excluded with a wash solution (10% (*v*/*v*) acetic acid, 10% (*v*/*v*) methanol, and 80% (*v*/*v*) distilled water) until the gel context was clear. The molecular weights of protein bands were determined using a protein standard mixture (Bio-Rad, Broad Range, California, USA) in the Image lab 6.0 (Bio-Rad) software [42].

#### 2.3.10. Heterocyclic Aromatic Amine Analysis

The HAAs extraction was done according to Messner and Murkovic’s method [43] with some modifications [44]. In summary, the cooked meat sample (1 g) and NaOH (1 M, 12 mL) were mixed at room temperature for 1 h and stirred with the packaging material (10 g, Extrelut NT, Merck, Darmstadt, Germany). For the extraction, ethyl acetate (75 mL), hydrochloric acid (0.1 N, 2 mL), and methanol (2 mL) were passed through Oasis cartridges (3 cc/60 mg, Oasis MCX, Waters). The analytes were eluted using a methanol:ammonia (19:1) solution. The samples were held at −18 °C before HPLC analysis, and the contents of the vial were dried in a 50 °C oven 1 day before HPLC analysis. The HPLC analysis began with adding 100 µL of methanol containing an internal standard (4,7,8-TriMeIQx) to a fully dried vial.

The HAA content of the beef steaks was determined using a reverse-phase analytical column (AcclaimTM 120 C18, 3 µm, 4.6 × 150 mm, Tosoh Bioscience GmbH, Stuttgart, Germany) in an HPLC (Thermo Ultimate 3000, Thermo Scientific, Waltham, MA, USA) with Diode Array Detector (DAD-3000). The HAAs were separated in a column oven at 35 °C with a flow rate of 0.7 mL/min. By running HAA mix stock solutions at different concentrations (10, 7.5, 5, 2.5, and 1 ng/g), the concentrations of HAA in the samples were measured. The external calibration curve approach was used for quantitative determination. In all of the evaluated HAA, the regression coefficients of HAA standard curves were greater than 0.999.

#### 2.3.11. Statistical Analysis

The current study was conducted in three replications using a completely randomized design, with each repeat being conducted in two parallels. The collected data were subjected to variance analysis using the Statistical Package for the Social Sciences (SPSS) statistical package program. The Duncan Multiple Comparison Test (*p* < 0.05) was used to calculate the average values of the significant sources of variation.

## 3. Results and Discussion

### 3.1. Some Chemical and Physicochemical Properties of the Raw Material

Table 1 shows the results of some chemical and physicochemical properties of beef steak meat. In studies on beef, researchers found similar findings. According to Oz [45], the water content of raw beef chops (M. *Longissimus dorsi*) ranged between 74.69–75.51%, with pH values ranging between 5.57 to 5.61. Oz and Zikirov [46], on the other hand, found that the water content of raw beef chops (M. *Longissimus dorsi*) was 72.54%, and the pH value was 5.56 in their analysis to evaluate the impact of the sous-vide cooking method on various quality parameters of beef. The TBARS value of raw beef chops was 0.477 mg MDA/kg, according to Zikirov [47], with L* values ranging from 40.57 to 42.89, a* values ranging from 19.12 to 26.76, and b* values ranging from 8.55 to 14.62. Another research found that the L* value of raw beef steak was 39.69, the a* value was 13.00, and the b* value was 2.05 [48]. The differences between the findings of the current study and those of the previous studies are thought to be due to differences in muscle, slaughter time, animal species, race, age, etc.

### 3.2. Water Contents of the Cooked Steaks

The fact that the cooking process decreases the water content of meat and meat products is explained by the shrinkage of myofibrillar proteins and perimysial connective tissue, caused by the high temperatures reached during the cooking process [49]. Table 2 shows the water content of the cooked samples. Cooked steaks had a water content ranging from 61.68 to 64.02%. The water content of cooked meat and meat products is affected by many factors [15]. In the present study, it is thought that the lack of effect of the marination process on a steak’s water content is related to the effect of cooking temperature used in the cooking process on water content rather than the effect of other factors on water content.

Some studies have reported that meat immersed in acidic marinades, causing a pH below 5.0, has a higher water content, less cooking loss, and is softer. The pH values of steaks cooked after marinating with various types of vinegar were between 5.60–5.68 in this analysis, with no statistically significant difference. Due to the marination conditions (marination type and period, etc.) using various vinegar types and the high pH value as a result of the cooking process after the marination process, it is assumed that there was no significant difference in the water content of the samples. In a study to assess the microbiological and sensory quality criteria of retail beef parts treated with acetic acid and lactic acid solutions, Kotula and Thelappurate [50] discovered that treatment with both acid solutions had no significant impact on the water content of the samples.

### 3.3. Cooking Loss Values of Steaks

Cooking loss in meat and meat products is a significant parameter that has a direct impact on both the consumer and the producer in terms of nutritional value and cost. Table 2 also shows the cooking loss values of the samples. It was discovered in this study that the cooking loss values of the samples coincided with the water content of the samples. The cooking loss values for the control group beef steaks and marinated beef steaks were determined to range between 46.52 and 49.18%. Following a statistical analysis, it was determined that there was no significant difference in cooking loss values across all samples. The probable reason why the marination process using different types of vinegar did not have a significant effect on the cooking loss of the samples was the destruction of the meat structure by the cooking process. This evaluation was also supported by the fact that the marination process using different vinegar varieties did not affect the water content and texture profile of the cooked samples. Similar findings have been reported in the literature. Mikel et al. [51] found that adding a 2% lactic acid and acetic acid mixture (*v*/*v*) to the bovine ribs loin did not affect the cooking loss of the samples. They also found that cooking losses ranged from 16.88 to 31.43% for samples with a thickness of 2.54 cm cooked with the grill method up to 66 °C internal temperature. Seong et al. [52] reported that the cooking losses of the samples ranged from 38.98 to 44.32% in their analysis to determine the effect of marination using different commercial vinegar on various quality characteristics of beef.

There are also studies showing that the marinating process has different effects on the cooking loss values of the samples. In some studies, the marination process was reported to increase cooking loss values, while in others the marination process decreased cooking loss values. In their analysis to determine the impact of marinades with different calcium concentrations on various quality criteria of chicken breast, Young and Lyon [53] confirmed that the marination process had caused an increase in cooking loss in samples. In another study, Barbanti and Pasquini [54] reported that marinating chicken breast meats with water, salt, sugar mixture, wheat flour, and milk protein, subjected to the tumbling process, baked at different temperatures (130–170 °C) and durations (from 4 to 12 min), caused an increase in cooking loss. Cesur [5] found the marination process resulted in a statistically significant increase (27.15–30.14%) in a cooking loss in chicken breast meats cooked after being marinated with cherry, pomegranate, orange, grape, and apple juice as compared to the control group samples (22.20%). The marinating process increased the cooking loss of the samples, according to the researcher, and the water in the meat passes into the marinade due to the high solid content in fruit juices (14%). Similarly, Koeipudsa et al. [55], in their study to increase the tenderness of chicken breast meat using alkaline or acidic solutions, determined that the cooking loss increased significantly with sodium bicarbonate (0.10, 0.15, 0.20, and 0.25 M) and lactic acid (0.05, 0.10, 0.15, and 0.20 M) marination compared to the control, and this was attributed to the presence of over-marinated solutions in the meat. On the other hand, Serdaroglu et al. [7] examined the impact of the marination process with different concentrations of citric acid (0.05 M, 0.1 M, and 0.2 M) and grapefruit juice (50% and 100%) on the quality of turkey breast meat. They found that citric acid application reduces or increases cooking loss depending on the concentration while marinating with grapefruit juice reduces cooking loss, but marinating with 100% grapefruit juice causes a significant reduction in cooking loss. The effect of marination with weak organic acids and salts on various quality parameters of beef *Longissimus dorsi* muscle was investigated by Aktas and Kaya [56]. They marinated their samples in plastic bags with lactic acid and citric acid solutions (0.5, 1, and 1.5%), NaCl solution (2, 4, and 6%), and CaCl_2_ solution (50 mM, 100 mM, and 150 mM) in a 1:1 ratio at 4 °C for 24 h, then baked them at 90 °C for 2 h. Cooking after marinating with NaCl, CaCl_2_, and citric acid solutions increased cooking loss values compared to control group samples, whereas cooking after marination with lactic acid solutions resulted in a decrease in cooking loss values compared to the control group samples, according to the researchers. Burke and Monahan [57] investigated the different quality parameters of beef treated with organic acids (acetic acid, citric acid, and lactic acid) as well as citrus juices (lemon juice and orange juice). They stated that the marinating process reduced cooking losses, but there was no statistically significant difference between the cooking losses of the marinated samples, which were below 35%.

The mainspring of cooking loss in meat and meat products is water loss, which is said to be caused by the temperature reached during cooking [58]. However, it is recognized that water loss is not the only cause of cooking loss; the process also extracts various water-soluble compounds (various proteins, salt, lipids, polyphosphates, etc.) [49,59].

### 3.4. pH Values of Steaks

One of the most important parameters influencing the quality of foodstuffs is the pH value. In this study, the pH values of the vinegars used in marinating beef were determined to be 3.18 in balsamic vinegar, 3.06 in pomegranate vinegar, 3.28 in apple vinegar, and 3.16 in grape vinegar. The use of undiluted vinegar in marination deteriorated the general structure of steaks according to preliminary studies, even if the marination was for a short period (approximately 1 h). Due to the high acidity of the vinegar used for marinating, the acidity of the vinegar was diluted to 0.5% during marinating to prevent deterioration of the general structure of the steaks, and the marination time was set at 4 °C for 2 h. As a result of dilution, the pH values of the vinegar used in marination were found to be 3.35 in balsamic vinegar, 3.25 in pomegranate vinegar, 3.40 in apple vinegar, and 3.32 in grape vinegar.

The release of basic character imidazole, sulfhydryl, and hydroxyl groups during the cooking process explains the increased pH value of meat and meat products [60]. Table 3 also shows the pH values of the cooked samples. Compared to the control group steaks, the marination procedure using various vinegar types resulted in a statistically significant decrease in the pH value of cooked samples. The use of vinegar in marinating beef steaks lowers the pH of the samples, which is due to the denaturation of the acetic acid present in the vinegar on the surfaces of the meat samples [61]. The low pH values of the vinegar varieties used in marination and the fact that the steaks absorb a certain amount of vinegar after marination are thought to be the reasons for the decrease in the pH value of beef steaks compared to the control group samples concerning the marination process with vinegar. The marinade absorption values of the samples demonstrate this finding. Immersion of meat in acidic marinades is stated to induce marinade absorption between muscle fibers, which speeds up muscle fiber swelling and proteolytic weakening [62]. Other researchers have discovered that the marination process with organic acids causes a significant decrease in the pH value of the meat [63,64,65,66,67,68]. According to Jones et al. [61], adding vinegar to a salted/dried meat product (biltong) consumed in South Africa reduced the pH value from 5.64 to 4.91. In the study conducted by Seong et al. [52] to determine the effect of the marination process using different commercial vinegars on various quality characteristics of beef, it was found that the pH value of raw meat varying between 5.51 and 5.60 decreased and varied between 5.11 and 5.37 after the marination process with vinegar. Yusop et al. [69] similarly found no significant difference in the pH values of chicken breast meats marinated for different periods (30–180 min) using marinades with different pH values in their research (pH 3–4.2).

### 3.5. TBARS Values of Steaks

Lipid oxidation is significant not only because it affects the quality of fatty foods but also because the resulting intermediates and major products are hazardous to human health. Table 3 also shows the TBARS values of the samples. As seen in the table, marination with various vinegar types increased the TBARS value compared to the control group samples. Still, there was no statistically significant difference between the control group steaks and the steaks cooked after marination with various vinegar varieties. The TBARS values of the samples ranged from 0.271 to 0.353 mg MDA/kg. Many scholars have come to similar conclusions. Hence, Mikel et al. [51] examined the effect of a 2% lactic acid and acetic acid mixture (*v*/*v*) applied to beef loin on sample storage for 112 days. They found that the TBARS value (3.41 mg MDA/g) in acid-treated cooked meats was higher than the TBARS value (3.34 mg MDA/g) in control group samples, but the difference was not statistically significant.

Cooking meat damages cell structure. Prooxidant interactions with polyunsaturated fatty acids may lead to lipid oxidation [70]. Furthermore, the high temperature reached during the cooking process reduces the activation energy required for lipid oxidation [71]. When different concentrations of lutein (100, 200 μg/g), sesame oil (250, 500 μg/g), ellagic acid (300, 600 μg/g), and olive leaf (100, 200 μg/g) extract were applied to pork meat, Hayes et al. [72] found that the TBARS content was lower than the control group samples. However, some studies indicate that the cooking process does not affect oxidation [73,74,75,76]. The reaction of reactive compounds determined by TBARS with various compounds found in meat, such as protein and amino acids, explains this result [47,49,77].

According to the literature, the panelists’ threshold for detecting oxidized taste-aroma in beef is 2 mg/kg [78]. As a result, the TBARS values of both non-marinated control group samples and marinated samples using various vinegar varieties were found to be far below this threshold value in this research.

### 3.6. Color Values of Steak

Color is a significant criterion that represents product quality parameters. Table 3 also shows the color values of the samples. The use of various vinegar types in the marination of steaks had different effects on the color values of the cooked samples in this study. The cooked samples marinated with pomegranate vinegar had the lowest L*, a*, and b* values. The cooked samples marinated with apple vinegar had the highest L*, a*, and b* values. This finding demonstrates that steaks marinated in pomegranate vinegar are darker, while steaks marinated in apple cider vinegar are lighter.

The L* value, which was found as 31.67 in non-marinated raw steaks, decreased in all samples (not marinated and marinated using different vinegar types) as a result of the cooking process and was found as 26.07 in control group samples and 22.23–30.37 in marinated samples using different types of vinegar in this research. Other researchers have determined that the L* value of beef subjected to dry heat application is reduced compared to raw samples (the meats are darker in color) [47,79,80,81]. This situation has been attributed to the Maillard reaction occurring during the process [80]. In the present study, statistically, the lowest L* value was found in the cooked samples marinated with pomegranate vinegar, the highest L* value was found in the cooked samples marinated with apple vinegar, and there was no statistically significant difference between the control group and the L* values of the samples marinated with balsamic and grape vinegar.

The a* value, which was found as 16.42 in non-marinated raw steaks, decreased in all samples (not marinated and marinated using different vinegar types) as a result of the cooking process, was 7.23 in the control group samples and ranged between 5.39–7.47 in the samples marinated using various vinegar styles in the present study. Other researchers also reported that the a* value of beef that was subjected to dry heat application reduced as compared to raw samples and that the decrease in the a* values of the meats as a result of the cooking process was due to pigment (oxymyoglobin) denaturation [47,49,79,80,81,82]. The current research found that the statistically lowest a* value was in the cooked samples marinated with pomegranate vinegar and the highest a* value was in the cooked samples in apple vinegar, but the a* value of this group was not significantly different from the a* value of the control group samples.

The b* value, which was found as 6.17 in non-marinated raw steaks, increased in all samples but one (the sample cooked in marinated with pomegranate vinegar) due to the cooking process. This value was 8.04 in the control group samples and ranged between 6.13–9.66 in the samples marinated with various vinegar types. Other researchers reported that the b* value of beef that was heat-treated increased relative to raw samples and that the increase in the b* values of the meats as a result of the cooking process was due to metmyoglobin formation and denaturation [47,83]. The statistically lowest b* value was found in the cooked marinated with pomegranate vinegar, and the statistically highest b* value was found in the cooked samples marinated with apple vinegar, with no significant differences found between the b* values of the other samples.

The statistically lowest L* value was found in the cooked samples by marinating with pomegranate vinegar in this study. Although the pH values of marinated and cooked steaks do not vary statistically, it is assumed that the pH value of pomegranate vinegar used in marination (3.25) was significantly lower than the pH values of other vinegar. Since using a low pH marinade causes denaturation of muscle proteins, the water binding capacity, and ability to reflect the light of the samples change, resulting in lighter-colored muscle formation [69,84]. In the current research, on the other hand, the statistically highest L* value was discovered in the cooked samples marinated with apple vinegar. It is assumed that apple vinegar’s pH (3.40) was significantly higher than other vinegar pH values might have influenced this result. Since using a high-pH marinade allows muscle proteins to swell, the light reflections of the samples change, resulting in a darker appearance [69]. The L* and b* values of the samples, on the other hand, were found to be in balance, and the highest color values (L* and b*) were found in the samples marinated with apple vinegar. The color of steaks is affected differently by various vinegar varieties; it is assumed that the extraction of various pigments in vinegar to the meat surface during marination was related to factors such as pH value, color, and content of vinegar [50,85]. Other researchers have reported similar findings [50,62,86].

There are also studies in the literature that show that the application of acid has no significant effect on the color of meat. Hence, Mikel et al. [51] found that the color values (L*, a*, and b*) of the cooked samples were unaffected by a mixture of 2% lactic acid, and acetic acid (*v*/*v*) applied to striploin. Similarly, Tänavots et al. [62] found that the pH of the marinade had no impact on the L*, a*, and b* values of cooked meat samples in their research.

### 3.7. Texture Profile Analysis Results of Steaks

Texture is one of the most significant quality measures of meat and meat products for consumers. The number of myofibrillar proteins, collagen, fat, and fiber in heat-treated products varies depending on the cooking conditions [87]. Table 3 shows the texture profile analysis results of the cooked samples. The marination process with various vinegar types had a significant (*p* < 0.05) effect only on the samples’ springiness value but not on their hardness, cohesiveness, or chewiness (*p* > 0.05). There is also research indicating that using organic acid has no significant effect on the texture properties of samples. Mikel et al. [51] reported that treating beef striploin with a mixture of 2% lactic acid and acetic acid (*v*/*v*) had no significant effect on the juiciness, brittleness, shear force calculation, or collagen content. On the other hand, there are also studies in the literature showing that acid application significantly affects the textural values of the samples [7]. The variations between the studies are thought to be due to differences in animal type, meat type, marinating conditions (marinade type, marination type, temperature, size, and so on), and cooking conditions (cooking method, cooking temperature, cooking time, etc.).

The hardness values of the steaks ranged from 137.26 to 237.90 N, and the hardness values of the samples were not statistically different. Furthermore, the springiness values of the steaks varied within narrow limits, ranging from 0.80 to 0.89 mm. The lowest springiness values belonged to steaks marinated with balsamic and grape vinegar; the highest springiness values belonged to control group steaks and steaks marinated with pomegranate vinegar. The cohesiveness values of the steaks ranged from 0.73 to 0.80, while the chewiness values ranged from 84.75 to 161.06 N.mm; however, the cohesiveness and chewiness values of the samples were not statistically different.

In some studies, it has been reported in the literature [6,57,88] that meat immersed in acidic marinades and thus having pH below 5.0 consequently absorbs water, has less cooking losses, and is less firm than the control group meat. Although the pH values of steaks cooked after marinating with various vinegar types (5.60–5.68) were significantly lower than the pH value of the cooked control group steaks (5.90), the pH values of all samples were found to be above 5.0 in this study. Due to the marination conditions (marination type and length, etc.) using various vinegar types and the high pH value resulting from the cooking process after the marination process, it is assumed that there is no significant difference in the hardness values of the samples. Furthermore, it is stated in the literature that one of the important factors influencing the hardness value of meat samples is the water content and the water loss seen in the samples [89]. It was found in this study that the water content, cooking loss, and hardness values of the steaks were all similar and that the marination process using different vinegar types had no significant impact on the water content, cooking loss, or hardness values of the steaks (*p* > 0.05).

According to the report, the tenderizing effect of acidic marinades is caused by two separate mechanisms. The first of these mechanisms, swelling of muscle fibers and connective tissue, dilutes the amount of load-resistant material so that tenderizing and swelling reach their maximum under the same conditions. Another mechanism may be attributed to cathepsins, according to the researchers. The optimum pH for cathepsins is between 3.5 and 5.0, and lowering the pH of the meat in acidic marination may significantly increase the proteolytic effect of cathepsins [57]. On the other hand, it is noted that when meat is immersed in acid solutions, the product may be harder due to the removal of the broth [90].

Contrary to the expectation that acidic marinades weaken the structure of meat by swelling, cathepsins contribute more to proteolysis in meat and the conversion of collagen to gelatin during cooking at low pH. It was found that the marinating process with different types of vinegar did not significantly affect the other textural properties of cooked steaks except springiness. Similarly, Aktas and Kaya [56] stated they could not find a correlation between penetrometer values and citric acid treatment and that the immersion process should be extended to achieve the tenderizing effect. In their investigation of the impact of acetic acid, citric acid, and lactic acid on the friability of reconstituted beef, Arganosa and Marriott [84] found that these three acids had a significant effect on both the total collagen content and collagen solubility of restructured meats.

Contrary to the various studies given above, the fact that there is no significant difference between the other textural parameters (hardness, cohesiveness, and chewiness) of the steaks marinated using different vinegars and cooked on a hot plate with a surface temperature of 200 °C, except for the springiness values, can be attributed to the use of vinegar in the marination process after dilution and the shortening of the marination period.

### 3.8. Changes in Myofibrillar Proteins (SDS-PAGE Profile)

SDS-PAGE profiles of myofibrillar proteins of steaks marinated with different vinegar types (balsamic, pomegranate, apple, and grape) and cooked on a hot plate are shown in Figure 1, Figure 2 and Figure 3. Based on the gel electrophoretograms obtained, the treatment samples are generally thought to have similar myofibrillar protein profiles. In other words, the band profiles of unmarinated/cooked steaks and steaks marinated/cooked with different vinegar types, designated as the control group, were similar. It is thought this might be related to the fact that the marination period is not long enough to cause changes in myofibrillar proteins, and/or the cooking process on myofibrillar proteins is more effective than marination application.

Proteins with molecular weights of 56, 43 (actin), 39, and 37 kDa were found to have high band density in all treatment groups; however, band density of proteins with molecular weights of 36, 27, 26, 19, 17, and 14.4 kDa was found to be lower than the other bands. Actin and myosin, which constitute the basic structure of thin and thick filaments, constitute an important part of myofibrillar proteins [91]. Myosin heavy chain of 200 kDa was not found in any of the samples in this study, but a protein band of 43 kDa, predicted to be actin, was observed in all of them. In the present study, although a similar SDS-PAGE profile was generally observed among the treatment group samples, it was determined that the band densities of some proteins differ depending on the marination application and the marinade type. The protein band density of 43 kDa, estimated to be actin, was lower in the samples marinated with pomegranate and grape vinegar in Trial I and samples marinated with pomegranate vinegar in Trials II and III compared to other treatment samples. The observed decreases in band density are thought to be linked to the observed protein aggregation associated with a high denaturation rate. It has been stated that aggregation prevents protein clusters from entering the gel pores, resulting in a decrease in band density [92]. On the other hand, compared to other treatment samples, the band density of the 17-kDa protein molecule was higher in pomegranate-marinated samples in Trial I, pomegranate- and apple-marinated samples in Trial II, and balsamic- and pomegranate-marinated samples in Trial III. This demonstrates that the experimental (animal) difference can affect some protein bands as well. On the other hand, small myofibrillar protein fractions can form due to myosin degradation, according to Dai et al. [93].

### 3.9. Heterocyclic Aromatic Amine (HAA) Contents of Steaks

The recovery of HAAs was determined using the standard addition method (10 ng/g, 7.5 ng/g, 5 ng/g, 2.5 ng/g, and 1 ng/g), and the recoveries were between 81.52–95.64%. The LOD (limit of detection = 3) and LOQ (limit of quantification = 10) values were between 0.004–0.025 and 0.013–0.085, respectively. These method validation parameters are consistent with data in the literature [43,94].

IQx compound appears to be one of the least studied compounds in research on HAAs. In this study, no IQx compound was found in any sample. Similarly, some studies in the literature show that IQx compound could not be detected in a variety of heat-treated meat and meat products. Indeed, IQx compound could not be detected in pork meats cooked at 98 °C for 1 h after marinating with sugar (1%) and soy sauce (5, 10, and 20%) by Lan and Chen (2002) [95], in beef chops cooked at 75–95 °C for 120–2400 min using the sous vide method by Oz and Zikirov [46], in beef chops cooked on a hot plate at 150–200 °C for 9 min with the addition of different molecular weights (low and medium) and amounts of chitosan (0.25–1%) by Oz et al. [18], and in beef meats cooked in a pan at 150 °C for 10 min by adding hawthorn extract at different levels (0, 0.5 and 1%) by Tengilimoglu-Metin et al. [33].

In the present study, IQ was determined in only one control group steak as 0.51 ng/g. No IQ compound was detected in other samples. Similarly, there are studies in the literature showing that IQ compound could not be determined in various cooked meat and meat products. Knize et al. [96] reported that IQ compound could not be detected in chicken breast meats cooked at 100 °C internal temperature after 4 h of marination with olive oil, lemon juice, apple cider vinegar, brown sugar, garlic, salt, and mustard, Toribio et al. [97] stated that IQ compound could not be found in beef steaks cooked in a Teflon pan for 4 min at 180–210 °C, Oz and Zikirov [46] reported that IQ compound could not be determined in beef chops cooked in a pan at 75 °C for 10 min, Oz et al. [18] stated that IQ compound could not be identified in beef chops cooked on a hot plate at 150–250 °C for 9 min with the addition of different molecular weights (low and medium) and amounts (0.25–1%) of chitosan, Tengilimoglu-Metin et al. [33] stated that IQ compound could not be detected in beef meat cooked in a pan at 150 °C for 10 min by adding hawthorn extract in different proportions (0, 0.5, and 1%), and Jinap et al. [98] reported that IQ compound could not be detected in beef meats that were kept in the microwave for 30 s after marinating with various marinades (cumin, shallots, coriander, lemongrass, turmeric, sugar, salt, and palm oil), and fried in palm oil at 160 °C for medium and well. On the other hand, there are studies in the literature showing that different amounts of IQ compound are detected in heat-treated meat and meat products. As a matter of fact, Tengilimoglu-Metin and Kizil [35] determined IQ compound varying between nd–0.66 ng/g in beef meat cooked in a pan and oven at 150–250 °C by adding artichoke extract in different proportions (0.5 and 1%), Tengilimoglu-Metin et al. [33] found IQ varying between 0.31–1.85 ng/g in beef meat cooked in a pan at 200–250 °C for 10 min by adding hawthorn extract at different rates (0, 0.5 and 1%), Szterk [99] determined IQ varying between 0.39 to 2.01 ng/g in beef tenderloins and steaks cooked on the grill at 180 °C (10 min) and 280 °C (7 min).

MeIQx compound is one of the most frequently studied and determined HAA compounds in cooked meat and meat products. In the present study, varying levels of MeIQx compound (0.59–2.22 ng/g) were detected in all steaks (Table 4). In addition, MeIQx was the only contributor to total HAAs in all marinated samples.

Different amounts of MeIQx compound have been found in various cooked meat and meat products, according to studies in the literature. Indeed, Gibis and Weiss [100] found MeIQx compound varying between 0.2–1.7 ng/g in beef patties marinated with marsh-mallow (*Hibiscus sabdariffa*) extract in different proportions (0.2, 0.4, 0.6, 0.8 g/100 g) and cooked for different times (120–180 s) on the grill with a surface temperature of 230 °C, Knize et al. [96], in their study, determined MeIQx compound up to 0.87 ng/g in ready-to-eat tenderloin and up to 0.89 ng/g in hamburgers, Tengilimoglu-Metin and Kizil [35] found MeIQx compound varying between nd–2.20 ng/g in beef meat cooked in a pan and oven at 150–250 °C by adding artichoke extract in different proportions (0, 0.5, and 1%), Iwasaki et al. [101] detected MeIQx compound up to 4.86 ng/g in beef meat cooked very well (until the internal temperature reaches 90 °C) after marination with garlic, onion, salt, and black pepper, Quelhas et al. [30] found MeIQx compund varying between 3.6–4.9 ng/g in pan-fried beef at 180–200 °C after marination process (1–6 h) with green tea, Knize et al. [96] identified MeIQx compound up to 6.1 ng/g in chicken breast meat cooked in different cooking times (14, 20, and 26 min) at 100 °C on the grill after 4 h of marination process with olive oil, lemon juice, apple cider vinegar, brown sugar, garlic, salt and mustard, Jinap et al. [98] found MeIQx compound up to 15.60 ng/g in barbecue medium-cooked beef and up to 15.12 ng/g in well-cooked beef after marinating using various marinades (cumin, shallots, coriander, lemongrass, turmeric, sugar, salt, and palm oil), Smith et al. [102] determined the MeIQx compound up to 30.3 ng/g in beef steaks cooked on the grill for 10 min at a surface temperature of 204 °C, which were marinated for 1 h with water, soybean oil, and vinegar.

In the present study, marinating with different vinegar types showed different effects on the MeIQx content of the samples. In this context, it was determined that there was no statistical difference between the MeIQx contents of the samples marinated with balsamic, apple, and grape vinegar and the MeIQx content of the control group samples. However, it was found that marinating with pomegranate vinegar resulted in increased MeIQx content in cooked steaks. It is thought that this result may be related to the content of substances with antioxidant properties in the vinegars and the change in these substances as a result of heat treatment. As is known, vinegar is rich in phenolic compounds associated with their high antioxidant capacity, and the bioactive properties of vinegar varieties vary widely depending on the raw material properties [103,104]. In a study investigating the total phenolic content of different vinegar varieties, the total phenolic content of pomegranate vinegar was found to be considerably higher than the phenolic content of apple and grape vinegar [12]. In the current study, it is thought that the higher total MeIQx content in steaks marinated with pomegranate vinegar compared to other samples may be due to the prooxidant effect of phenolic substances in pomegranate vinegar under cooking conditions. Although it is known that vinegars have an antioxidant effect due to the phenolic compounds they contain [11], it is also known that antioxidant substances can have a prooxidant effect depending on the concentration, structure, and substrate as a result of heat treatment [24,31,105,106,107]. Similarly, Johansson and Jägerstad [108] declared that the antioxidant or prooxidant effect was strongly associated with concentration. On the other hand, Gibis and Weiss [109] reported that the changing variation of precursors also affects the formation of HAAs. Since HAAs are formed in the presence of creatine, free amino acids, and sugar which is frequently used in marinating mixtures [110], it is thought that the differences observed in the sugar content of vinegar varieties in the current study may also affect this result.

MeIQ compound was not detected in any of the steaks. Similarly, studies show that MeIQ compound could not be determined in cooked meat and meat products. Indeed, Toribio et al. [97] reported that MeIQ compound could not be detected in beef steaks cooked in a Teflon pan for 4 min at 180–210 °C. Jinap et al. [98] stated that MeIQ compound could not be identified in beef meats that were kept in the microwave for 30 s after marinating with various marinades (cumin, shallots, coriander, lemongrass, turmeric, sugar, salt, and palm oil) and medium-fried in palm oil at 160 °C. Tengilimoglu-Metin et al. [33] reported that MeIQ compound could not be found in beef meat cooked in a pan at 150 °C for 10 min by adding hawthorn extract in different proportions (0, 0.5, and 1%).

Compound 7,8-DiMeIQx was not detected in any of the steaks. Similarly, studies show that 7,8-DiMeIQx compound could not be determined in heat-treated meat and meat products. Indeed, Toribio et al. [97] reported that 7,8-DiMeIQx compound could not be detected in beef steaks cooked in a Teflon pan for 4 min at 180–210 °C, Lee et al. [111] stated that 7,8-DiMeIQx compound might not be found in beef steaks cooked for 10 min at a surface temperature of 200 °C in a Teflon electric pan with the addition of different amounts (0–10 g) of extra virgin olive oil, Jinap et al. [98] reported that compound 7,8-DiMeIQx could not be identified in beef cooked after marinating with various marinades (cumin, shallots, coriander, lemongrass, turmeric, sugar, salt, and palm oil), Oz and Zikirov [46] stated that 7,8-DiMeIQx compound could not be determined in beef chops cooked in a pan at 75 °C for 10 min, Oz et al. [112] reported that 7,8-DiMeIQx compound could not be found in beef chops cooked on a hot plate at 150 °C for 9 min by adding conjugated linoleic acid at different rates (0–0.5%), Jamali et al. [32] reported that 7,8-DiMeIQx compound could not be detected in beef meatballs fried in deep fat (with soybean oil) at 160–220 °C by adding rosehip extract (0.1%).

Compound 4,8-DiMeIQx was not detected in any of the steaks. Similarly, there are studies in the literature showing that 4,8-DiMeIQx compound could not be determined in marinated and heat-treated meat and meat products. As a matter of fact, Gibis and Weiss [109] reported 4,8-DiMeIQx compound could not be detected in beef meatballs marinated with marsh-mallow (Hibiscus sabdariffa) extract in different proportions (0.2, 0.4, 0.6, 0.8 g/100 g) and cooked for different times (120–180 s) on the grill with a surface temperature of 230 °C, Lee et al. [111] stated that 4,8-DiMeIQx compound could not be found in beef steaks cooked for 10 min at a surface temperature of 200 °C in a Teflon electric pan with the addition of different amounts (0–10 g) of extra virgin olive oil, Viegas et al. [113] reported that 4,8-DiMeIQx compound could not be identified in beef meat cooked on barbecue with coconut shell and wood at 200 °C, Jinap et al. [98] stated that 4,8-DiMeIQx compound could not be found in the beef they cooked after the marination process they performed using various marinades (cumin, shallots, coriander, lemongrass, turmeric, sugar, salt, and palm oil), Oz and Zikirov [46] reported that 4,8-DiMeIQx compound could not be determined in beef chops cooked in a pan at 75 °C for 10 min, Oz et al. [114] stated that 4,8-DiMeIQx compound could not be identified in beef chops cooked for 9 min on a hot plate at 150–250 °C by adding conjugated linoleic acid in different proportions (0–0.5%), Tengilimoglu-Metin et al. [33] stated that 4,8-DiMeIQx compound could not be determined in beef meat cooked in a pan at 200–250 °C for 10 min with different proportions (0, 0.5 and 1%) of hawthorn extract.

PhIP compound was not detected in any of the steaks. Similarly, there are studies in the literature showing that PhIP compound could not be determined in heat-treated meat and meat products. Indeed, Iwasaki et al. [101] reported that PhIP compound could not be detected in beef cooked with garlic, onion, salt, and pepper after marinating with different cooking temperatures (low, medium, and well) in the pan and grill, Jinap et al. [98] stated that PhIP compound could not be found in the beef they cooked after marinating using various marinades (cumin, shallots, coriander, lemongrass, turmeric, sugar, salt, and palm oil), Oz and Zikirov [46] reported that PhIP compound could not be found in beef chops cooked at 95 °C with the sous vide method for 120–240 min

AαC compound was not detected in any of the steaks. Similarly, there are studies in the literature showing that AαC compound could not be determined in heat-treated meat and meat products. Indeed, Knize et al. [96] stated that AαC compound could not be detected in ready-to-eat tenderloins, Oz and Zikirov [46] stated that AαC compound could not be determined in beef chops cooked at 95 °C with the sous vide method for 120–240 min. Tengilimoglu-Metin et al. [33] reported that AαC compound could not be found in beef meat cooked in a pan at 150 °C for 10 min by adding hawthorn extract in different proportions (0, 0.5, and 1%). Tengilimoglu-Metin and Kizil [35] stated that AαC compound could not be detected in beef meat cooked in a pan and oven at 150–200 °C by adding artichoke extract in different proportions (0, 0.5, and 1%).

MeAαC compound was not detected in any of the steaks. Similarly, studies show that MeAαC compound could not be determined in heat-treated meat and meat products. Indeed, Oz et al. [112] reported that MeAαC compound could not be detected in beef cutlets cooked on a heating plate at 150–250 °C for 9 min by adding conjugated linoleic acid in different proportions (0–0.5%), Oz et al. [18] stated that MeAαC compound could not be found in beef cutlets cooked on a heating plate at 150–250 °C for 9 min with the addition of different molecular weights (low and medium) and amounts (0.25–1%) of chitosan, Tengilimoglu-Metin and Kızıl [35] reported that by adding artichoke extract in different proportions (0, 0.5, and 1%), MeAαC compound could not be determined in beef meat cooked in a pan and oven at 150–250 °C.

Table 4 also shows the total HAA content of the steaks. As can be seen from the table, the total HAA content of the samples varied between 0.59–2.22 ng/g. The individual HAAs contributing to the total HAA content of the control group were IQx and MeIQx. In contrast, the total HAA content of the samples marinated with different vinegars consisted of only MeIQx. It was found that there was no statistical difference between the total HAA contents of the samples marinated with balsamic, apple, and grape vinegar and that of the control group samples. On the other hand, it was determined that samples marinated with pomegranate vinegar had a higher amount of total HAA than other samples. This result showed that the effect of vinegar marination on HAA formation in beef steaks depended on the vinegar type. These differences observed among vinegar varieties are thought to be related to the number of phenolic substances with antioxidant properties and the reactions of these substances to heat treatment, as we explained in detail in the individual MeIQx content section. Various studies show that additives rich in antioxidant components increase HAA formation in various meat products [36,106,107].

Kilic et al. [106] declared that turmeric had a dose-related inhibition on the amount of HAAs in cooked chicken meatballs. While the strongest inhibitory effect was determined in 0.5% meatballs with turmeric added, a 1.0% turmeric addition caused an increase in total HAA content.

Uzun and Oz [36] reported that the total HAA content varied between nd–1.61 ng/g in their study examining the HAA (IQx, IQ, MeIQx, MeIQ, 7,8-DiMeIQx, 4,8-DiMeIQx, PhIP, AαC, and MeAαC) formation in beef patties produced by adding basil in different proportions (0, 0.25, 0.50, 0.75, and 1%) and cooked at different temperatures (150, 200, and 250 °C). The researchers found that, depending on the usage rate and cooking temperature, using basil in meatball production had both a decreasing and increasing impact on total HAA content. While the inhibitory effect of basil on the formation of HAA has been attributed to the antioxidant effect of the phenolic compounds possessed by basil, its enhancing effect on the formation of HAA has been attributed to the prooxidant effect of the phenolic compounds.

Oz and Zikirov [46] in their study to determine the effect of different cooking methods (sous-vide, boiling, and stir-frying) on HAA (IQx, IQ, MeIQx, MeIQ, 7,8-DiMeIQx, 4,8-DiMeIQx, PhIP, AαC, and MeAαC) formation in beef chops, as in this study, examined 9 individual HAA formations. The total HAA content in the samples studied ranged between 0.032–0.940 ng/g, according to the researchers.

In their study to determine the impact of chitosan use on HAA formation i beef chops, Oz et al. [18] cooked their samples for a total of 9 min on a hot plate with a surface temperature of 150, 200, and 250 °C. The researchers determined that in the samples they analyzed, the total HAA (IQx, IQ, MeIQx, MeIQ, 7,8-DiMeIQx, 4,8-DiMeIQx, PhIP, AαC, and MeAαC) content varied between nd–4.21 ng/g. The total HAA content in the control group samples prepared without adding chitosan increased with the increase in cooking temperature. The addition of chitosan reduced the total HAA content (14.3–100%) at all cooking temperatures. Researchers attributed the inhibitory effect of chitosan to the formation of total HAA to the high-water holding capacity of chitosan.

In their study, Iwasaki et al. [101] determined the effect on the formation of HAA by cooking beef marinated using garlic, onion, salt, and black pepper at different cooking temperatures (low, medium, and well) in a pan and grill and they reported that the total HAA (MeIQx, 4,8-DiMeIQx, and PhIP) content of the samples varied between nd–1185.5 ng/100 g and the marination process reduced the HAA content of the samples.

Gibis and Weiss [34] stated that the overall HAA (MeIQx, 4,8-DiMeIQx, PhIP, Harman, and Norharman) content varied between 5.36–17.75 ng/g in meatball samples cooked for 160 s on the grill with a surface temperature of 220 °C in their analysis to determine the effect of adding cellulose fiber at different rates (0.5, 1, 2, and 3%) to beef on HAA formation. The number of oligosaccharides in the cellulose fiber had an inhibitory and enhancing effect on HAA formation, according to the researchers. They also found that cellulose fibers’ high water holding capacity had a direct impact on HAA content.

Due to factors such as the impact of many factors on the formation of HAA, the variety of vinegar used in the current study, and the use of only vinegar in the marination process, comparing the HAA results obtained in the current study with data in the literature is very difficult. On the other hand, it is possible to say that the total HAA content obtained in the present study is low compared to the data in the literature. It is known that many factors, such as meat type, amount of HAA precursors, the preparation method of the meat product, marination process, marinating method, marinating used in marination, effects of marinades, heat treatment conditions, etc., affect the content of HAA in heat-treated meat and meat products [19,22,105,115,116]. Some inconsistency between the literature data and the findings obtained in this study is thought to be due to these reasons.

## 4. Conclusions

The present study results illustrate that vinegar use in beef steak marination affects meat quality depending on the vinegar variety used. The lowest and highest L*, a*, and b* values in the study were detected in steaks cooked after marination with pomegranate and apple cider vinegar, respectively. The marination process using different types of vinegar had a significant effect (*p* < 0.05) only on the springiness value of the textural properties of the steaks, and the use of balsamic and grape vinegar in the marinating process decreased the springiness value compared to the steaks of the control group. The myofibrillar proteins of the treatment samples generally had a similar band profile. Our results showed that the effect of vinegar marination on HAA formation in beef steaks depended on the vinegar type. Among the vinegar varieties examined, only pomegranate vinegar and marinating process caused an increase in the total HAA content of beef steaks. The daily intake of HAAs per person has been reported to be 0–15 µg/day [117]. In the present study, the intake level (0.222 µg) remains well below the established limit even when consuming 100 g of steaks marinated with pomegranate vinegar, which has the highest total HAA content. However, to better understand the effect of vinegar type on HAA formation in the marinating process, it is thought that there is a need for new studies investigating the bioactive substance content and chemical composition of vinegar varieties.

## Figures and Tables

**Figure 1 foods-11-03251-f001:**
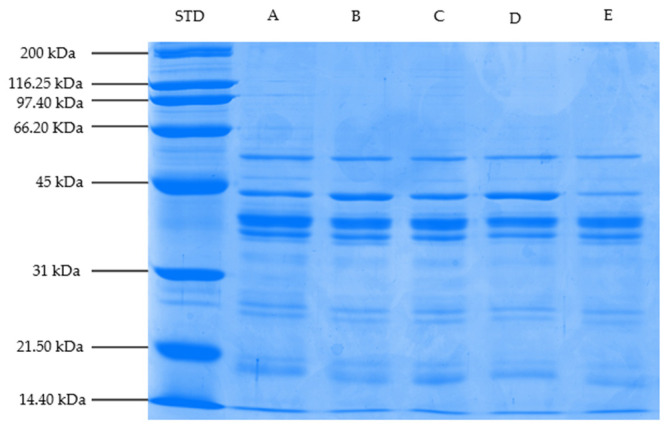
SDS-PAGE profile of myofibrillar proteins of treatment samples (Block–I) (STD: Protein standard, A: Control (Not Marinated), B: Balsamic, C: Pomegranate, D: Apple, and E: Samples of cooked steak marinated in Grape Vinegar).

**Figure 2 foods-11-03251-f002:**
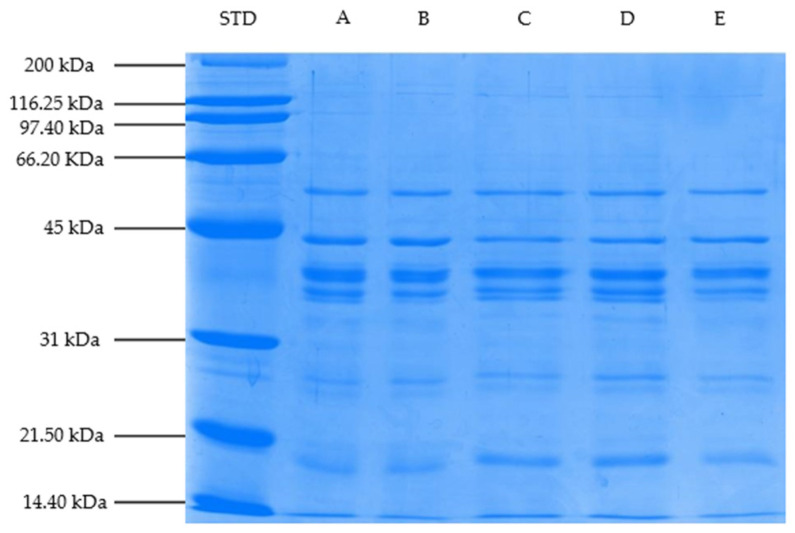
SDS-PAGE profile of myofibrillar proteins of treatment samples (Block–II) (STD: Protein standard, A: Control (Not Marinated), B: Balsamic, C: Pomegranate, D: Apple, and E: Samples of cooked steak marinated in Grape Vinegar).

**Figure 3 foods-11-03251-f003:**
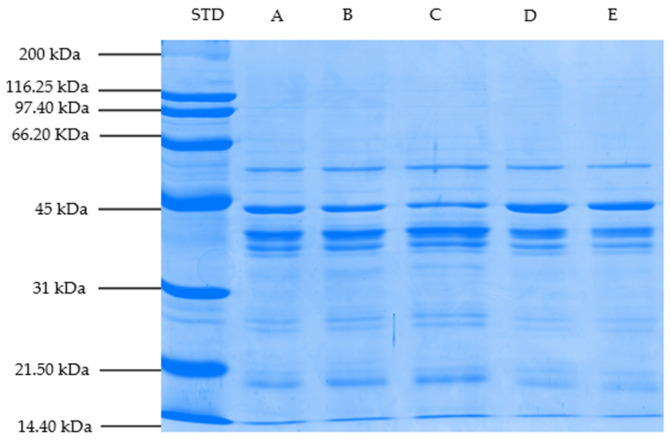
SDS-PAGE profile of myofibrillar proteins of treatment samples (Block–III) (STD: Protein standard, A: Control (Not Marinated), B: Balsamic, C: Pomegranate, D: Apple, and E: Samples of cooked steak marinated in Grape Vinegar).

**Table 1 foods-11-03251-t001:** The results of some chemical and physicochemical properties of raw beef steak meat (*n* = 6).

Analysis	Results
Water (%)	78.09 ± 0.33
pH	5.76 ± 0.09
TBARS (mg MDA/kg)	0.181 ± 0.05
L*	31.67 ± 1.52
a*	16.42 ± 0.96
b*	6.17 ± 0.54

**Table 2 foods-11-03251-t002:** Some chemical and physicochemical analysis results of the cooked samples (mean ± SD) (*n* = 6).

Treatment	Water (%)	Cooking Loss (%)	pH	TBARS	L*	a*	b*
Control	62.41 ± 2.02 a	46.83 ± 13.48 a	5.90 ± 0.06 a	0.271 ± 0.06 a	26.07 ± 2.33 b	7.23 ± 0.68 ab	8.04 ± 0.82 b
Balsamic	64.02 ± 1.93 a	46.52 ± 11.81 a	5.60 ± 0.11 b	0.353 ± 0.05 a	24.41 ± 1.67 b	6.54 ± 0.60 b	7.46 ± 0.76 b
Pomegranate	61.68 ± 3.34 a	49.18 ± 11.29 a	5.67 ± 0.07 b	0.326 ± 0.10 a	22.23 ± 1.64 c	5.39 ± 0.46 c	6.13 ± 0.80 c
Apple	63.26 ± 3.83 a	46.90 ± 12.12 a	5.67 ± 0.04 b	0.294 ± 0.08 a	30.37 ± 3.38 a	7.47 ± 0.41 a	9.66 ± 0.78 a
Grape	62.29 ± 3.84 a	47.60 ± 13.66 a	5.68 ± 0.08 b	0.275 ± 0.19 a	25.16 ± 1.80 b	7.26 ± 0.67 ab	8.04 ± 0.59 b
Sign	ns	ns	**	ns	**	**	**

Different letters (a–c) in the same column are significantly different (*p* < 0.05). SD: Standard deviation; ns: not significant (*p* > 0.05). **: *p* < 0.01.

**Table 3 foods-11-03251-t003:** The texture profile analysis results of the cooked samples (mean ± SD) (*n* = 6).

Treatment	Hardness (N)	Springiness (mm)	Cohesiveness	Chewiness (N.mm)
Control	215.37 ± 140.74 a	0.89 ± 0.08 a	0.79 ± 0.10 a	157.77 ± 120.28 a
Balsamic	216.35 ± 55.61 a	0.80 ± 0.03 b	0.74 ± 0.03 a	128.43 ± 36.62 a
Pomegranate	211.28 ± 109.02 a	0.89 ± 0.08 a	0.80 ± 0.11 a	157.61 ± 107.95 a
Apple	237.90 ± 65.65 a	0.87 ± 0.07 ab	0.78 ± 0.04 a	161.06 ± 45.17 a
Grape	137.26 ± 84.37 a	0.80 ± 0.06 b	0.73 ± 0.09 a	84.75 ± 65.64 a
Sign	ns	*	ns	ns

Different letters (a–b) in the same column are significantly different (*p* < 0.05). SD: Standard deviation; ns: not significant (*p* > 0.05). *: *p* < 0.05.

**Table 4 foods-11-03251-t004:** MeIQx and total HAA contents of the steaks (mean ± SD) (*n* = 6).

Treatment	MeIQx (ng/g)	Total HAAs (ng/g)
Control	0.95 ± 0.62 b	1.04 ± 0.71 b
Balsamic	0.67 ± 0.50 b	0.67 ± 0.50 b
Pomegranate	2.22 ± 1.14 a	2.22 ± 1.11 a
Apple	0.59 ± 0.84 b	0.59 ± 0.85 b
Grape	1.34 ± 1.00 ab	1.34 ± 1.00 ab
Sign	*	*

Different letters (a–b) in the same column are significantly different (*p* < 0.05). SD: Standard deviation. *: *p* < 0.05.

## Data Availability

The data presented in this study are available on request from thecorresponding author.

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
