# Peer review of "The Effects of the Marination Process with Different Vinegar Varieties on Various Quality Criteria and Heterocyclic Aromatic Amine Formation in Beef Steak"

_foods, 2022, doi:10.3390/foods11203251_

Round 1
Reviewer 1 Report
Article Foods-1961179: ‘The Effects of Marination Process with Different Vinegar Varieties on Various Quality Criteria and Heterocyclic Aromatic Amine Formation in Beef Steak’ by: Halenur FencioÄŸlu, Emel Oz, Sadettin Turhan, Charalampos Proestos, Fatih Oz
The authors propose to study how different vinegar varieties (balsamic, pomegranate, apple, and grape) affected HAA formation and other meat quality parameters. I found the objectives of the work original and interesting and worthy of being researched. In my opinion, the research was well conducted, but the material and methods need complementary information. The results and the discussion are presented appropriately but, In my opinion, the discussion regarding HAA is very extensive, more than 400 lines, which makes it difficult for the reader to pay attention. This discussion could be more succinct, particularly in HAAs that were not found in the present study.
Abstract
Line 15 - ‘3,12 – 4,13%; replace by ‘3.12 – 4.13%’
Line 17 – ‘p>0,05’; replace by ‘p>0.05’. Made similar corrections in all the abstract
Line 19- ‘L*, a*, and b*’; remove italic (also Line 151).
Introduction
Line 72 – ‘changes in protein content’; I think the authors want to refer to qualitative rather than quantitative changes in proteins, so eliminate the word ‘content’.
Materials and Methods
Line 106 – the meat samples used are poorly characterized ‘beef steak meat was obtained from three different carcasses’; did the authors use undifferentiated meat in their work? for research purposes, the name of the muscle used should be referred to. The authors also do not mention the postmortem time of the meat used, age of animals and sex, which may also affect several of the parameters studied, namely TBARS, color, and texture.
2.2.1. Marinating Process – as the vinegar was diluted to 0.5% acidity with purified water (10x dilution), the control steaks are also marinated with water ?. Are the steaks wiped with paper before weighing? Please clarify in the text. this may explain why the controller lost weight (table 2).
2.2.2. Cooking Process – please indicate the thickness of the steaks.
2.2.6. pH Analysis – are made in raw meat? please clarify in the text.
Indicate in all methods that the analysis was made on raw, cooked or on both raw and cooked.
2.2.7. Lipid Oxidation -Please indicate the percentage of solutions used, namely for the trichloroacetic acid solution; thiobarbituric acid solution; I think the authors used a standard curve; what substance was used (TMP; TEP, malonaldehyde)? briefly mention it.
Line 142 – ’12 ml’; replace by ’12 mL’; please made similar corrections in all the text.
Line 145 - ‘blind sample’; do you mean ‘blank sample’?
2.2.8. Color Analysis – In raw meat, the color parameters are very dependent on the time of contact with the air (blooming). The color was measured in a fresh cut or after air exposure to air (for how long) ?
2.2.9. Texture Profile Analysis - Please indicate sample thickness, fiber orientation, and some test parameters, such as deformation percentage and probe speed.
2.2.12. Statistical Analysis – please refer to the level of significance used with Duncan test
What do you mean by ‘three replications’’? 3 samples for each treatment? Or 3 replications of 2 samples = 6 samples (n) as mentioned in the tables?
Results and Discussion
Tables - Is redundant in all the tables a column with the ‘n’. Put the ‘n’ in the Title of the table, for example ‘Table 1. The results of some chemical and physicochemical properties of raw beef steak meat (n=6).
In all Tables add this information like this ‘Means with different superscripts within the same row are significantly different at p<0.05. Values are represented as means ± standard deviation. ‘Confirm the p value and if is the standard deviation.
Line 444 – ‘samples' elasticity’ the authors are referring to the springiness? If so please write elasticity (springiness) for better clarification. The same for ‘ internal stickiness,’ ‘internal stickiness (cohesiveness).
Table 4 In material and methods the texture parameters referred to are ‘The hardness, springiness, cohesiveness, and chewiness parameters of the samples were determined’ but in the results, springiness isn’t present and is resilience (I think that resilience and springiness are different calculations by the software).
Also, I think that resilience is dimensionless.
3.10. Heterocyclic Aromatic Amine (HAA) Contents of Steaks
In my opinion, the discussion regarding HAA is very extensive, more than 400 lines, which makes it difficult for the reader to pay attention. This discussion could be more succinct, particularly in HAAs that were not found in the present study.
Author Response
Response Letters to the Reviewers
Foods
foods-1961179
Title: The Effects of Marination Process with Different Vinegar Varieties on Various Quality Criteria and Heterocyclic Aromatic Amine Formation in Beef Steak
Dear Respected Editor and Reviewers,
Thanks a bunch for your positive and constructive comments on our manuscript entitled “The Effects of Marination Process with Different Vinegar Varieties on Various Quality Criteria and Heterocyclic Aromatic Amine Formation in Beef Steak” (ID: foods-1961179). The comments are very valuable and helpful for revising and improving the quality of our manuscript. We have carefully revised the manuscript based upon your concerns and comments. All amendments made on the revised manuscript are marked. The detailed information of the responses to the comments is presented as follows.
We do hope that the revised manuscript will be acceptable for publication in Foods. If there are any further comments/suggestions, please inform me any time. Thank you again for your kind help.
Sincerely yours,
Prof. Dr. Fatih Oz
Response to the Reviewers' comments:
Reviewer #1:
Comments to the Author
The authors propose to study how different vinegar varieties (balsamic, pomegranate, apple, and grape) affected HAA formation and other meat quality parameters. I found the objectives of the work original and interesting and worthy of being researched. In my opinion, the research was well conducted, but the material and methods need complementary information. The results and the discussion are presented appropriately but, In my opinion, the discussion regarding HAA is very extensive, more than 400 lines, which makes it difficult for the reader to pay attention. This discussion could be more succinct, particularly in HAAs that were not found in the present study. So, I found the article worthy to be published after major revision.
Thank you for valuable comments on our manuscript. We would like to show our heartfelt thanks to Reviewer 1 for his/her kind and helpful comments. We would also like to thank the referee for deeming our research worthy of publication. According to his/her advice, we have amended the relevant parts of the manuscript. Below are our answers to each of Reviewer 1's questions and comments.
Abstract
Line 15 - ‘3,12 – 4,13%; replace by ‘3.12 – 4.13%’
Thank you for your suggestion. The suggested correction has been made throughout the revised manuscript.
Line 17 – ‘p>0,05’; replace by ‘p>0.05’. Made similar corrections in all the abstract
Thank you for your suggestion. The suggested correction has been made throughout the revised manuscript.
Line 19- ‘L*, a*, and b*’; remove italic (also Line 151).
Thank you for your suggestion. The suggested corrections have been made throughout the revised manuscript.
Introduction
Line 72 – ‘changes in protein content’; I think the authors want to refer to qualitative rather than quantitative changes in proteins, so eliminate the word ‘content’.
Thank you for your nice comment. You are right. The word “content” has been removed from the sentence.
Materials and Methods
Line 106 – the meat samples used are poorly characterized ‘beef steak meat was obtained from three different carcasses’; did the authors use undifferentiated meat in their work? for research purposes, the name of the muscle used should be referred to. The authors also do not mention the postmortem time of the meat used, age of animals and sex, which may also affect several of the parameters studied, namely TBARS, color, and texture.
Thank you for your comment. The raw materials used in the present study were the meat that had completed the rigor stages and was rested for 24 h after slaughter. In addition, the beef steaks were provided from three male beef (Simmental) which were two years old. The analyzes were performed on a total of 6 samples, in two parallels, from steaks obtained from three different male cattle. The related parts of the manuscript have been revised.
2.2.1. Marinating Process – as the vinegar was diluted to 0.5% acidity with purified water (10x dilution), the control steaks are also marinated with water ?. Are the steaks wiped with paper before weighing? Please clarify in the text. this may explain why the controller lost weight (table 2).
Thank you for your nice comment. We had forgotten to add this information to the manuscript. The sliced beef steaks were separated to 5 pieces after being got into the same dimensions on average and were put into special bags (Elektrola, EVP-SV2230). A randomly selected group was not marinated (even if water) and this group was evaluated as the control group. To the other 4 groups, the mentioned forms of vinegar (200 mL) were added. Then, all samples including the control group were kept at 4°C for 2 h. Thus, the marinated samples were marinated according to the dipping (static) marination process. In addition, the steaks wiped with paper before weighing. The related part of the manuscript has been revised.
2.2.2. Cooking Process – please indicate the thickness of the steaks.
Thank you for your suggestion. The thickness of the steaks was 2 cm as we mentioned in the material section.
2.2.6. pH Analysis – are made in raw meat? please clarify in the text. Indicate in all methods that the analysis was made on raw, cooked or on both raw and cooked.
Thanks for your attention. pH analysis was performed on both vinegar samples and steaks (raw and cooked steaks). This information has been added to the revised manuscript. In addition, the same information was provided for the other analyzes as suggested.
2.2.7. Lipid Oxidation -Please indicate the percentage of solutions used, namely for the trichloroacetic acid solution; thiobarbituric acid solution; I think the authors used a standard curve; what substance was used (TMP; TEP, malonaldehyde)? briefly mention it.
Thanks for your comment. TEP was used for the calculation (standard curve). Required information regarding TBARS analysis has been added to the revised manuscript.
Line 142 – ’12 ml’; replace by ’12 mL’; please made similar corrections in all the text.
Thank you. The corrections have been made throughout the revised manuscript.
Line 145 - ‘blind sample’; do you mean ‘blank sample’?
Yes. Thank you. The correction has been made.
2.2.8. Color Analysis – In raw meat, the color parameters are very dependent on the time of contact with the air (blooming). The color was measured in a fresh cut or after air exposure to air (for how long) ?
Thank you for your good question. The beef we used as raw material in the present study was the meat that had completed the rigor mortis stages and was rested for 24 h after slaughter. For this reason, it is considered that the meats have completed the first blooming (opening event). On the other hand, the meats brought under the cold chain from the slaughterhouse to the laboratory were kept at room temperature for 30 min, and then analyzes/processes were started. The related parts of the manuscript have been revised.
2.2.9. Texture Profile Analysis - Please indicate sample thickness, fiber orientation, and some test parameters, such as deformation percentage and probe speed.
Thank you for your comment. The detailed information has been added to the revised manuscript.
2.2.12. Statistical Analysis – please refer to the level of significance used with Duncan test
Thank you. The level of significance (p<0.05) used with Duncan test has been added.
What do you mean by ‘three replications’’? 3 samples for each treatment? Or 3 replications of 2 samples = 6 samples (n) as mentioned in the tables?
Thanks for your inquiry. As you know, meat is a non-homogeneous food item. For this reason, three different treatments were performed to increase the accuracy of the analyses and the reproducibility of the results, and in addition, each treatment was repeated in two parallels. Thus, each result was expressed as the mean of 6 analysis samples.
Results and Discussion
Tables - Is redundant in all the tables a column with the ‘n’. Put the ‘n’ in the Title of the table, for example ‘Table 1. The results of some chemical and physicochemical properties of raw beef steak meat (n=6).
Thank you for your sensible suggestion. The relevant corrections have been made in the title of the tables.
In all Tables add this information like this ‘Means with different superscripts within the same row are significantly different at p<0.05. Values are represented as means ± standard deviation. ‘Confirm the p value and if is the standard deviation.
Thank you. The relevant corrections have been made in the tables.
Line 444 – ‘samples' elasticity’ the authors are referring to the springiness? If so please write elasticity (springiness) for better clarification. The same for ‘ internal stickiness,’ ‘internal stickiness (cohesiveness).
Yes, you are completely right. Sorry for that. These corrections have been made throughout the revised manuscript.
Table 4 In material and methods the texture parameters referred to are ‘The hardness, springiness, cohesiveness, and chewiness parameters of the samples were determined’ but in the results, springiness isn’t present and is resilience (I think that resilience and springiness are different calculations by the software). Also, I think that resilience is dimensionless.
Yes, you are completely right. Sorry for that. These corrections have been made throughout the revised manuscript.
3.10. Heterocyclic Aromatic Amine (HAA) Contents of Steaks
In my opinion, the discussion regarding HAA is very extensive, more than 400 lines, which makes it difficult for the reader to pay attention. This discussion could be more succinct, particularly in HAAs that were not found in the present study.
Thank you for your comment. Relevant parts of the manuscript have been simplified in line with your recommendation. In addition, some information about the possible reasons for the results has been added to the revised manuscript.

Reviewer 2 Report
This study evaluated different marination vinegar on the quality and safety of beef steaks. It is of pratical importance and presents some interesting results. I think the paper could be further considered, but only after addressing the following comments:
I would suggest to use “.” rather than “,” for decimals.
For non-significant comparisons, don’t use expressions “lower”, “higher” etc. In many cases, the deviation is quite large, the larger or smaller mean values is occurring by chance.
The length of Section 3.10 can be significantly reduced. Since only two HAAs (IQ and MeIQx) could be detected, discussion of the others can be combined.
In the current form, the discussion is a bit too descriptive. Authors could try to explore the mechanism of the results, for example, why pomegranate vinegar increased total HAAs.
Author Response
Response Letters to the Reviewers
Foods
foods-1961179
Title: The Effects of Marination Process with Different Vinegar Varieties on Various Quality Criteria and Heterocyclic Aromatic Amine Formation in Beef Steak
Dear Respected Editor and Reviewers,
Thanks a bunch for your positive and constructive comments on our manuscript entitled “The Effects of Marination Process with Different Vinegar Varieties on Various Quality Criteria and Heterocyclic Aromatic Amine Formation in Beef Steak” (ID: foods-1961179). The comments are very valuable and helpful for revising and improving the quality of our manuscript. We have carefully revised the manuscript based upon your concerns and comments. All amendments made on the revised manuscript are marked. The detailed information of the responses to the comments is presented as follows.
We do hope that the revised manuscript will be acceptable for publication in Foods. If there are any further comments/suggestions, please inform me any time. Thank you again for your kind help.
Sincerely yours,
Prof. Dr. Fatih Oz
Response to the Reviewers' comments:
Reviewer #2:
Comments to the Author
This study evaluated different marination vinegar on the quality and safety of beef steaks. It is of pratical importance and presents some interesting results. I think the paper could be further considered, but only after addressing the following comments:
Thank you for valuable comments on our manuscript. We would like to show our heartfelt thanks to Reviewer 2 for his/her kind and helpful comments. According to his/her advice, we have amended the relevant parts of the manuscript. Below are our answers to each of Reviewer 2's questions and comments.
I would suggest to use “.” rather than “,” for decimals.
Thank you for your comment. “ . (dot)” has been used instead of “ , (comma)” in decimal expressions. Please check the revised manuscript.
For non-significant comparisons, don’t use expressions “lower”, “higher” etc. In many cases, the deviation is quite large, the larger or smaller mean values is occurring by chance.
Thank you for your sensible comment. You are completely right. The sentences including these statements have been removed from the revised manuscript.
The length of Section 3.10 can be significantly reduced. Since only two HAAs (IQ and MeIQx) could be detected, discussion of the others can be combined.
Thank you for your comment. The discussion part in the section 3.10 has been designed and written in accordance with the purpose of the present research. Although only two of the HAAs analyzed in the present study were identified, it is considered to be important in terms of comparing the results in the literature in the discussion part with the results obtained in the current study. However, this part has been reduced in line with your recommendation.
In the current form, the discussion is a bit too descriptive. Authors could try to explore the mechanism of the results, for example, why pomegranate vinegar increased total HAAs.
Thank you for your good comment. Discussion parts of the manuscript have been simplified in line with your recommendation. In addition, some information about the possible reasons for the results has been added to the revised manuscript.

Reviewer 3 Report
The investigators did a LOT of work by varying vinegar type in a so-called marinade (water + vinegar, 0.5% acidity) and detecting its effects on several aspects of cooked steak. Although I found this study is interesting at the beginning, overall the manuscript requires further modification.
1) M&M must provide more details for reproducibility.
2) Discussion is unnecessary long with no relevant point with the study. Please be kindly reminded that your objectives stated that the aim of this study was to "see how different vinegar varieties affected HAA formation". In this regard, I assumed that this also applied for other properties. In this case, your discussion should mainly focus on how different type of vinegar played role on each properties. Mostly, I could not find why or how there is no significant effects (or the vice versa) due to the different vinegar types.
Also, we must keep in mind that differences in marination process and cooking method affect such properties of meat. This means that you may want to provide such information of previous studies so that the readers have enough information.
3) Please be careful when interpret the data with no statistical difference.
Please find more detailed review in the attach file

Author Response
Response Letters to the Reviewers
Foods
foods-1961179
Title: The Effects of Marination Process with Different Vinegar Varieties on Various Quality Criteria and Heterocyclic Aromatic Amine Formation in Beef Steak
Dear Respected Editor and Reviewers,
Thanks a bunch for your positive and constructive comments on our manuscript entitled “The Effects of Marination Process with Different Vinegar Varieties on Various Quality Criteria and Heterocyclic Aromatic Amine Formation in Beef Steak” (ID: foods-1961179). The comments are very valuable and helpful for revising and improving the quality of our manuscript. We have carefully revised the manuscript based upon your concerns and comments. All amendments made on the revised manuscript are marked. The detailed information of the responses to the comments is presented as follows.
We do hope that the revised manuscript will be acceptable for publication in Foods. If there are any further comments/suggestions, please inform me any time. Thank you again for your kind help.
Sincerely yours,
Prof. Dr. Fatih Oz
Response to the Reviewers' comments:
Reviewer #3:
Comments to the Author
The investigators did a LOT of work by varying vinegar type in a so-called marinade (water + vinegar, 0.5% acidity) and detecting its effects on several aspects of cooked steak. Although I found this study is interesting at the beginning, overall the manuscript requires further modification.
Thank you for valuable comments on our manuscript. We would like to show our heartfelt thanks to Reviewer 3 for his/her kind and helpful comments. According to his/her advice, we have amended the relevant parts of the manuscript. Below are our answers to each of Reviewer 3's questions and comments.
1) M&M must provide more details for reproducibility.
Thank you for your good comment. Material and method section have been improved and more details about the analyses have been added to the text.
2) Discussion is unnecessary long with no relevant point with the study. Please be kindly reminded that your objectives stated that the aim of this study was to "see how different vinegar varieties affected HAA formation". In this regard, I assumed that this also applied for other properties. In this case, your discussion should mainly focus on how different type of vinegar played role on each properties. Mostly, I could not find why or how there is no significant effects (or the vice versa) due to the different vinegar types. Also, we must keep in mind that differences in marination process and cooking method affect such properties of meat. This means that you may want to provide such information of previous studies so that the readers have enough information.
Thank you for your valuable comments. You are completely right. The article has been revised in line with your suggestions. In addition, some information about how different type of vinegar affected the meat properties and HAA formation has been added to the revised manuscript.
3) Please be careful when interpret the data with no statistical difference.
Thank you for your comment. The sentences including these statements have been removed from the revised manuscript.
Please find more detailed review in the attach file
Thank you for your careful reviewing of our manuscript. The related parts you mentioned have been revised. Please check the revised manuscript. The answers for the questions in the pdf file are below;
Were the raw meat bloomed before the analysis? If yes, at what temperature and for how long?
Thank you for your good question. The beef we used as raw material in the present study was the meat that had completed the rigor mortis stages and was rested for 24 h after slaughter. For this reason, it is considered that the meats have completed the first blooming (opening event). On the other hand, the meats brought under the cold chain from the slaughterhouse to the laboratory were kept at room temperature for 30 min, and then analyzes/processes were started. The related parts of the manuscript have been revised.
About the cooked meat how many meat samples per treatment were used, what was the temperature of the meat?
Thank you for your nice comment. As mentioned in the Statistical Analysis section, the current study was conducted in three replications using the completely randomized design, with each repeat being conducted in two parallels. For color analysis of every meat piece, four readings (two of one side, two of other side) were done. As for the temperature, color analysis was performed after the temperature of all samples (raw and cooked) was brought to room temperature. This information has been added to the revised manuscript.
Did the investigators use homogenized meat? If not, please consider revising the detail of samples preparation (line 125).
Thank you for your caution. The color analysis was done after the cooked meats were cooled at room temperature and before homogenizing. The related part has been revised.
Were the samples cut or prepared into any shape before the analysis? What was the temperature of the samples? Please indicate the compression condition.
Thank you for your comment and questions. The part of the texture profile analysis including the answers of these questions has been revised.
Were the control sample left in the refrigerator without any marination or it was subjected to distilled water. Were the meat covered? In any case, this suggested that the investigators need further clarification in the M&M.
Thank you for your nice comment. We had forgotten to add this information to the manuscript. This is our mistake. The sliced beef steaks were separated to 5 pieces after being got into the same dimensions on average and were put into special bags (Elektrola, EVP-SV2230). A randomly selected group was not marinated (even if water) and this group was evaluated as the control group. To the other 4 groups, the mentioned forms of vinegar (200 mL) were added. Then, all samples including the control group were kept at 4°C for 2 h. Thus, the marinated samples were marinated according to the dipping (static) marination process. The related part of the manuscript has been revised.
Did the investigators determined pH of each vinegar before marination? Were their pH level comparable?
Thank you for your comment. Yes, we determined the pH values of the vinegars. As mentioned in the manuscript, pH values of the vinegar marinades used in the marination process were found to be 3.40 (±0.01a) for apple vinegar, 3.35 (±0.01b) for balsamic vinegar, 3.32 (±0.01c) for grape vinegar, and 3.25 (±0.01d) for pomegranate vinegar.
What does it mean for "different content of the vinegar types"?
Sorry for this sentence. Here, we mean that different vinegars have different ingredients. And we state that different contents of vinegars affect this result. However, this part of the manuscript has been removed from the revised manuscript.
Pleae elaborate on the differences between what?
Thank you for your comment. To be honest, it is quite difficult to know for certain which process caused the difference. Therefore, in this sentence, the potential reasons for the differences seen the related results determined in the present study with the data in the literature are tried to be explained. It is considered that these potential differences may be caused by marinade type, duration, application method, etc. However, this part of the manuscript has been removed from the revised manuscript.
Since statistics showed no significant differences among treatments, it's not appropriate to say which is higher than which. So, I'd strongly recommend to revise this sentences.
Thank you for your comment. The sentences including these statements have been removed from the revised manuscript.
Overall, your results indicated that no significant influences of vinegar types on water content and cook loss of the cooked beef -- This also need discussion on why.
Thank you for your good comment. Since heterocyclic aromatic amines are formed as a result of the cooking process and the samples need to be cooked to determine the cooking loss, all analyzes were carried out on the final ready-to-consumption samples, that is, the cooked samples. For this reason, although the use of vinegar had no any effect on the water content of the samples, we may not say that the vinegar type did not affect the water content and cooking loss of the samples, since the effect may have disappeared as a result of the cooking in the present study. However, we emphasized here that the amount of water removed from the meat as a result of the cooking may be a stronger factor on the water content of the samples compared to the water content, which changes due to the changes in the water holding capacity due to the pH change, and we supported this with the literature.
I am not sure the purpose of this paragraph. With marinade uptake, you show that your meat pick up some marinade. Also, you did not analyze anything in the marinade after marination.
Thank you. We had written this part to observe whether marinade absorption had an effect on water content or cooking loss. However, the use of different vinegar in the marination of the steaks had no any effect on water content and cooking loss value. On the other hand, this section (marinade absorption of steaks) has been removed from the revised article because the reviewers suggestions on the shortening of the discussion section.
In this study, you need a discussion why no effects of vinegar type on water content/marinade pickup, and compare your results with other studies.
Thank you for your comment. As in your question above, this part has been tried to be explained by specifying the literature since the analyzes were made on the cooked sample. As we mentioned before that this is could be due to the fact that the effect of the cooking process on the mentioned analyses (water content, cooking loss, etc.) is greater than the effect of using different vinegars in the marinade.
The statement on “Steaks with the highest water content and cooked after marinating in balsamic vinegar had the lowest cooking loss. It was discovered that steaks cooked after marinating in pomegranate vinegar and having the lowest water content had the highest cooking loss value.”: Again their is no signficant. So, it's not appropriate.
Thank you for your comment. You are right. The sentences including these statements have been removed from the revised manuscript.
The statement on “marinating process”: Please show what type of marination process each literature used; e.g., static marination, tumbling marination, and cooking method, and whether they look at the effect of vinegar type of what.
Thank you for your comment. We marinated our samples according to the dipping method (static). The related information has been added to the revised manuscript as you wish.
The statement on “control group”: define control
Thank you for your comment. Control group states unmarinated beef steak. But the control group beef steak was also kept at 4 C for 2 h like other marinated steak groups. The needed explanations have been added to the section of 2.3.1. Marinating Process.
The statement on “significant difference between the cooking losses of the marinated samples, which were below 35%.”: Based on overall lit review, what would be the reasons of no significance in cooking loss of your study?
Thanks for your attention. The probable reason why the marination process using different types of vinegar did not have a significant effect on the cooking loss of the samples is the destruction of the meat texture by the cooking process. In other words, it is considered that the cooking process had a greater effect than the marination process on these analyzes done in the samples. This evaluation was also supported by the fact that the marination process using different vinegar varieties had no effect on the water content and texture profile of the cooked samples. This information has been added to the revised manuscript.
Section 3.8. Texture Profile Analysis Results of Steaks: This part is great.
Thank you very much for your nice and valuable comment regarding this section. This is a pride for us.
Statement on honey word: I still do not know why or how IQx not found in this study.
Thank you for your nice questions. As you know and we mentioned in the text before, HAAs can be formed in proteinaceous foods such as meat during their cooking. It means that they may not form in all cooked meat, depending on the cooking conditions. Indeed, in the current research, only two HAAs of the nine HAAs studied could be detected. The possible reason for that may be due to the fact that the activation energy needed for the formation of other HAAs could not be reached. But, unfortunately, in this research, we could not study the activation energy. In addition, the formation mechanisms of the HAAs are very complex and a lot of factor can affect their formation in meat. Maillard reaction is very important in the formation of the HAAs in cooked meat. On the other hand, it is known that the Maillard reaction, which takes place during the cooking of complex foods such as meat, is more complex than other foodstuffs. At this stage, it is known that many intermediate components (eg pyridine, pyrazine, etc.) affect the type of HAA that can be formed. Therefore, the explanations about this issue are made in general rather than specificity in the literature on HAAs formation in meat. With the model system studies planned to be carried out in the future, this situation will be tried to be revealed more clearly.
Statement about the names of HAAs: define this term
Thank you for your caution. This is our mistake. We had forgotten to add this information. But, in the revised manuscript, we have added the section 2.2. Chemicals. Please check the revised manuscript.
Then, please provide the discussion specific for your study. I went to the end of the study but found nothing
Thank you for your comment. In this paragraph, we have mentioned that the comparison of the results obtained in the present study with those in the literature due to the differences in meat type, marination type and technique, cooking condition is very difficult. In addition, as known, meat is complex food and cooking process makes it very complex. However, in this paragraph, a general assessment (as we mentioned above) has been made that the results obtained in the current study is lower than the results in the literature. I hope you find the explanation we made here appropriate. In addition, some new comments have been added to the relevant section.

Round 2
Reviewer 1 Report
Dear authors
The authors made the suggested corrections.
Kind regards
Author Response
Response to the Reviewers' comments:
Reviewer #1:
Comments to the Author
Dear authors
The authors made the suggested corrections.
Kind regards
Yes, we have revised our manuscript according to the reviewers’ suggestion. We would like to show our heartfelt thanks to Reviewer 1 for deeming our research worthy of publication.

Reviewer 2 Report
The paper has been improved.
Author Response
Response to the Reviewers' comments:
Reviewer #2:
Comments to the Author
The paper has been improved.
Yes, we have revised our manuscript according to the reviewers’ suggestion. We would like to show our heartfelt thanks to Reviewer 2 for deeming our research worthy of publication.

Reviewer 3 Report
Quality of the revised manuscript is significantly improved. The investigators have shown adequate information to each comment.
As for cooking loss, I agree with the investigators that the effects of cooking might be overpower the effects of the main factors (vinegar types). Anyway, the higher water loss in treated samples compared to control might be because the meat pick up excessive water during marination and could not hold all of the pick-up fluid during cooking. On the other hand, the control lost only their original moisture during cooking. Some of the literature showed that if the meat was marinated with water, there would be more cooking loss than control and samples marinated with acidic or basic marinades. Please see the following article for more discussion.
Improving tenderness of breast meat of spent-laying hens using marination in alkaline or acidic solutions. Asia-Pacific Journal of Science and Technology: Volume: 24. Issue: 04. Article ID.: APST-24-04-08.
Author Response
Response to the Reviewers' comments:
Reviewer #3:
Comments to the Author
Quality of the revised manuscript is significantly improved. The investigators have shown adequate information to each comment.
Yes, we have revised our manuscript according to the reviewers’ suggestions. Thank you for valuable comments on our manuscript. We would like to show our heartfelt thanks to Reviewer 3 for deeming our research worthy of publication.
As for cooking loss, I agree with the investigators that the effects of cooking might be overpower the effects of the main factors (vinegar types). Anyway, the higher water loss in treated samples compared to control might be because the meat pick up excessive water during marination and could not hold all of the pick-up fluid during cooking. On the other hand, the control lost only their original moisture during cooking. Some of the literature showed that if the meat was marinated with water, there would be more cooking loss than control and samples marinated with acidic or basic marinades. Please see the following article for more discussion.
Improving tenderness of breast meat of spent-laying hens using marination in alkaline or acidic solutions. Asia-Pacific Journal of Science and Technology: Volume: 24. Issue: 04. Article ID.: APST-24-04-08.
Thank you for your nice comment. According to your advice, we have revised the related part of the manuscript and added this article to the text. Please see the revised manuscript.
